# Detection of Somatic Mutations with ddPCR from Liquid Biopsy of Colorectal Cancer Patients

**DOI:** 10.3390/genes12020289

**Published:** 2021-02-19

**Authors:** Uršula Prosenc Zmrzljak, Rok Košir, Zoran Krivokapić, Dragica Radojković, Aleksandra Nikolić

**Affiliations:** 1Molecular Biology Laboratory, BIA Separations CRO, Labena Ltd, 1000 Ljubljana, Slovenia; rok.kosir@labena.si; 2Department of Colorectal Diseases, Clinic for Digestive Surgery, First Surgical Clinic, Clinical Center of Serbia, 11000 Belgrade, Serbia; krivokapiczoran@gmail.com; 3Medical Faculty, University of Belgrade, 11000 Belgrade, Serbia; 4Serbian Academy of Sciences and Arts, 11000 Belgrade, Serbia; 5Institute of Molecular Genetics and Genetic Engineering, University of Belgrade, 11000 Belgrade, Serbia; dada@imgge.bg.ac.rs (D.R.); aleksni@imgge.bg.ac.rs (A.N.)

**Keywords:** colorectal cancer, liquid biopsy, cell-free DNA, somatic mutations, KRAS, BRAF, ddPCR, hemorrhoids

## Abstract

Liquid biopsy and cell-free DNA (cfDNA) show great promise in cancer diagnostics. In this study, we designed a custom droplet digital PCR (ddPCR) assay for the quantification and quality control of cfDNA isolated from serum. The assay was validated on a group of locally advanced colorectal cancer (CRC) patients and two control groups—patients with hemorrhoids and healthy individuals. The assay shows a high correlation with Qubit measurement (r = 0.976) but offers a higher dynamic range. Mean concentrations of cfDNA were 12.36 ng/µL, 5.17 ng/µL, and 0.29 ng/µL for CRC, hemorrhoid patients, and healthy controls, respectively. The quality of cfDNA was assessed with the measurement of B-cell DNA contamination. On a subset of CRC patients, we compared the mutation status on KRAS (G12A, G12D, G12V, G13D) and BRAF (V600E) genes in the primary tumor and cfDNA isolated from the serum. A total of 70.6% of primary tumor samples were mutated, and the mean fractional abundance of mutations was 9.50%. The matching serum samples were mutated in 38% cases with an average fractional abundance of 0.23%. We conclude that any decisions based solely on the amount of cfDNA present in patient serum must be interpreted carefully and in the context of co-morbidities. This study explores the potential of ddPCR somatic mutations detection from liquid biopsy as a supplement to tissue biopsy in targeted personalized CRC patient management.

## 1. Introduction

Colorectal cancer (CRC) remains one of the most common cancers worldwide, behind lung and breast cancer [1]. One of the options for disease management is molecular targeted therapies. In colorectal cancer, most biological drugs that are used target the epidermal growth factor receptor (EGFR), which mediates cellular response to growth signals and is known to be over-expressed in 50–80% of colorectal tumors [2,3,4]. Thus, EGFR represents a suitable monoclonal therapy target, but this kind of treatment is effective only in a subset of patients who do not harbor mutations in the mitogen activated protein kinase (MAPK) pathway [5,6]. Therefore, guidelines for the treatment of colorectal cancer patients recommend genetic testing of tumors in order to determine mutation status in RAS and RAF genes to select the patients suitable for the treatment [7].

The detection of somatic mutations has always represented a technological challenge because only a small subset of tumor cells harbors these mutations, and methods are often limited in sensitivity for their determination. The introduction of new techniques with improved sensitivity has enabled somatic mutation measurement in primary tumors, in metastases large enough to enable sampling, and lately also from liquid biopsies [8]. Droplet digital PCR (ddPCR) is one of the most sensitive methods for the detection of somatic mutations [9] and is currently one of the best methods used in the analysis of cfDNA [10].

cfDNA shows great potential as an alternative diagnostic and prognostic non-invasive biomarker that need to be validated for different purposes in different types of cancer. The analysis of cfDNA mutations has already been introduced in the guidelines of non-small cell lung cancer management where secondary EGFR mutation T790M is being followed for early detection of acquired resistance to first-line EGFR-tyrosine-kinase-inhibitors [11]. Another approach is being evaluated in metastatic melanoma patients where the amount of total cfDNA is used as a surrogate biomarker for tumor burden [12]. In this aspect, the possible contamination with DNA from lysed lymphocytes may contribute to falsely increased DNA levels [13]; therefore, appropriate quality control measures are required. Nevertheless, the colorectal cancer screening test based on the cfDNA methylation status of SEPTIN9 promotor is already FDA approved [14] and cfDNA, therefore, holds great potential in cancer diagnostics. Several papers describe the potential use of cfDNA in CRC management—as a marker of tumor postoperative recurrence in early-stage cancer [15,16,17] or as a tool for the assessment of tumor burden in metastatic patients [18,19]. Activating mutations in KRAS are the main reason for non-responsiveness to EGFR-targeting drugs among CRC patients. Hence, tracking the KRAS mutations can be used as a monitoring tool for primary or secondary resistance to cetuximab or panitumumab with the goal to allow early implementation of adaptive treatment strategies [20,21,22]. But most of all, higher survival rates could be achieved with better and more sensitive diagnostics of early-stage cancers that are still curable [23]. The level of concordance of cfDNA and tumor mutations greatly correlate with the disease stage [24]. If the early-stage patients are considered, then the combined testing is a safer option, where liquid biopsy is surrogate to tissue biopsy.

In this study, we present our ddPCR assay for quantification of cfDNA coupled with the B-cell DNA rearrangement assay for the assessment of blood contamination during sample preparation. We applied this assay to quantify cfDNA in colorectal cancer patients, patients with hemorrhoids, and healthy controls. In a subset of patients, we compared the mutation status of KRAS and BRAF genes in the primary tumor and serum samples. All our patients had locally advanced rectal carcinoma (LARC) and did not have any confirmed metastases at the time of clinical diagnosis. Our study represents one of a few that explores the potential of cfDNA as a supplement to tissue biopsies in non-metastatic patients.

## 2. Materials and Methods

### 2.1. Patients and Controls Selection

A total of 34 patients (32% female, 62% male) with primary colorectal cancer were included in the study. All patients were treated by the same surgical team following standard oncological radical procedures based on the location and the stage of the disease. All relevant clinical, demographic, and epidemiological data were collected for each patient. Two control groups were selected. One group consisted of 37 patients (35% female, 65% male) who underwent surgical removal of hemorrhoids. Relevant clinical data were collected and none of the participating patients had any malignancies at the time of blood draw. The second group included 13 healthy controls (77% female, 23% male) without any gut-related or any other severe health problems. All patients and controls were enrolled in this study with written informed consent between 2012 and 2015 and were treated at the First Surgical Clinic of Clinical Centre of Serbia after study approval by the Ethics Committee of the Clinical Centre of Serbia, approval no. 562/2.

### 2.2. DNA Isolation from Whole Blood Samples

Whole blood from healthy controls was drawn into EDTA-coated tubes and isolated on the same day with the use of an Extractme blood kit (Blirt, Gdansk, Poland) according to the manufacturer’s protocol. The concentration was measured spectrophotometrically (V-730 UV-Visible Spectrophotometer, JASCO, Tokyo, Japan) and was adjusted with nuclease-free water (Invitrogen, Carlsbad, CA, USA) to a final concentration of 1 µg/µl.

### 2.3. DNA Isolation from Tumor Samples

The isolation of DNA from fresh tumor samples was described previously [25]. Briefly, the tumor samples were collected at the time of surgical excision of the tumor. Genomic DNA was isolated with GeneJET genomic DNA purification kit (Thermo Fisher Scientific, Waltham, MA, USA). The concentration of isolated DNA was determined spectrophotometrically (V-730 UV-Visible Spectrophotometer, JASCO, Tokyo, Japan). Samples were diluted with nuclease-free water (Invitrogen, Carlsbad, CA, USA) to a final concentration of 1 µg/µl.

### 2.4. Cell-Free DNA Isolation from Serum Samples and Fluorometric Quantification

Blood samples were taken prior to surgery in tubes containing clot activator and centrifuged at 2000 g for 10 min after collection. After centrifugation, the serum was transferred to 1.5 mL tubes and stored at −80 °C until analysis. Isolation of cfDNA was performed with QIAamp MinElute cfDNA (Qiagen, Hilden, Germany) according to the manufacturer’s protocol. The volume of serum samples used for isolation varied from 200 µL to 1500 µL. All isolations were eluted in 50 µL of elution buffer from the kit. The concertation of cfDNA was first determined with Qubit dsDNA HS Assay Kit (Thermo Fisher Scientific, Waltham, MA, USA) according to the manufacturer’s protocol with a standard curve in concentration from 7.8 pg/µL to 500 pg/µL. The measurement was performed on NanoDrop 3300 Fluorospectrometer (NanoDrop, (Thermo Fisher Scientific, Waltham, MA, USA). The final cfDNA concentration in serum samples was calculated according to the initial volume of serum used for isolation.

### 2.5. ddPCR Assay for Quantification of Cell-Free DNA and Leukocyte Contamination Evaluation

For the purposes of ddPCR quantification of cfDNA, a genomic DNA assay was designed on chromosome 7 (GRCh38.p13: 5562562–5562812). The leukocyte contamination evaluation was performed with the assay targeted to the B-cell DNA rearrangement region described previously [13].

Each 20 µL ddPCR reaction consisted of 10 µL of ddPCR Supermix for Probes (Bio-Rad, Hercules, CA, USA), genomic DNA and B-cell rearrangement assays, and 4 µL of DNA.

Each reaction mixture was transferred to a cartridge along with 70 μL of Droplet Generation Oil (Bio-Rad, Hercules, CA, USA) and partitioned into droplets using the QX200 Droplet Generator (Bio-Rad, Hercules, CA, USA). Droplets were transferred to 96-well plates (Eppendorf, Hamburg, Germany,) and sealed using the PX-1 plate sealer (Bio-Rad, Hercules, CA, USA). The PCR was performed in a T100 Thermal Cycler (Bio-Rad, Hercules, CA, USA). The cycling conditions were as follows: 95 °C 10 min, 1 cycle; 94 °C 30 s, 55 °C 1 min, 40 cycles; 98 °C 10 min, 4 °C ∞. Following end-point amplification, the individual droplets were measured with the QX200 Droplet Reader (Bio-Rad, Hercules, CA, USA) and analyzed using QuantaSoft software (Bio-Rad, Hercules, CA, USA) to determine the target cDNA amount. All samples with less than 10.000 droplet counts were omitted from the following analysis.

### 2.6. Somatic Mutation Detection (ddPCR) in Tumor and Serum DNA Samples

The mutation status for KRAS p.G12A, KRAS p.G12D, KRAS p.G12V, KRAS p.G13D, and BRAF p.V600E was measured with the use of ddPCR mutation assays (BioRad, Hercules, CA, USA). Tumor samples were analyzed in one well per patient. Genomic DNA samples isolated from the whole blood of three individuals from the healthy control group were used as a negative control in tumor sample analysis. Prior to the ddPCR reaction, DNA samples were fragmented with restriction enzyme Hind III HF (New England Biolabs, Ipswich, MA, USA), which included 1 µL of restriction buffer (supplied with the enzyme), 2.5 µL 10-fold diluted enzyme, 500 ng DNA sample, and water to bring to 10 µL. The restriction mixture was incubated at 37 °C for 15 min. Immediately after restriction, a 20 µL ddPCR reaction was prepared. It consisted of 10 µL of ddPCR Supermix for Probes (Bio-Rad, Hercules, CA, USA), 1 µL of mutated (MUT) assay, 1 µL of wild-type (WT) assay, and 2 µL of restricted DNA. Serum samples were analyzed in three wells per patient. A pool of cfDNA isolated from the serum of patients with hemorrhoids without CRC in a similar concentration range as cfDNA of CRC patients was used as a negative control in serum sample analysis. Each 20 µL ddPCR reaction consisted of 10 µL of ddPCR Supermix for Probes (Bio-Rad, Hercules, CA, USA), 1 µL of MUT assay, 1 µL of WT assay, and 5 µL of DNA. Droplets were prepared as described previously.

### 2.7. Sanger Sequencing

PCR and sequencing reactions were described previously [26]. The cycling conditions for cold PCR were described previously [27] and were as follows: 95 °C 5 min, 1 cycle; 94 °C 1 min, 70 °C 1 min, 78 °C 1 min (sec target 79 °C, step size 0.1, step delay cycles 3), 57 °C 45 s, 60 °C 45 s, 30 cycles; 40 °C 10 s, 1 cycle. For sequencing, one positive sample (patient no. 17) with 20% of somatic KRAS G12V mutation was selected as the positive control, and a sample of one individual from the healthy control group was selected as the negative control.

### 2.8. Statistical Analysis

Statistical analysis was performed using R Statistical Software [28]. Differences in cfDNA concentration between groups were assessed using the Mann–Whitney U test. Pearson’s product-moment correlation coefficient was used for the comparison of Qubit and ddPCR assays of cfDNA concentration estimation. All graphical representations in figures were plotted using the visualization package for R ggplot2 [29].

For the assessment of B-cell DNA contamination, the percentage was calculated in the following manner: the number of B-cell DNA assay copies was divided by the number of genomic DNA assay copies. For each group, a mean value with a 95% confidence interval is reported. The difference between groups was assessed with the use of a one-way ANOVA test.

## 3. Results

### 3.1. Validation of the Assay for cfDNA Concentration Measurement

The ddPCR assay was designed in the intergenic region of chromosome 7 (GRCh38.p13: 5562562–5562812). To ensure that the performance of the assay would not be compromised by any genomic event, we performed a literature search to assess whether the region is a good candidate for quantification assay. We did not find any data indicating that this region is involved in any known translocations, duplications, or copy no. variation events, or that the sequence is present in any pseudogenes. No known SNPs (SNPCheck V3) or mutations (COSMIC) were found in the primer and probe annealing sites. For the purposes of determining the linearity of the assay, DNA isolated from the whole blood of healthy controls was prepared in five 10-fold dilution series starting with a concentration of 2.5 ng/µL. The assay shows linearity through the dilution series down to 0.06 copies per µL (Figure 1). To determine the limit of detection (LOD), one sample of isolated cfDNA from a hemorrhoid patient was diluted several times and measured in at least 16 replicas. The lowest dilution when all replicates were positive was 200-fold, which corresponds to a concentration of 0.006 ng/µL. We set this concentration as the limit of detection of our assay. 

Performance of the assay was correlated with fluorometric assay Qubit (r = 0.9764) (Figure 2A). The range of concentrations that can be measured with ddPCR is much wider compared to Qubit (Figure 2B). The samples that could be measured with both methods were used for the construction of a Bland–Altman plot (Figure 2C). No significant bias was found for either of the methods as evident in distance from the average of the difference between two paired measurements (distribution along the *y*-axis). The samples of different concentration ranges are equally measured with both methods (distribution along the *x*-axis).

### 3.2. Measurement of cfDNA Concentration and Quality Assessment with B-Cell Assay

The concentration of cfDNA in serum was measured with ddPCR in CRC patients, hemorrhoid patients, and healthy controls (Figure 3). The mean value for the CRC patients’ group was 12.36 ng/µL, for hemorrhoid patients 5.17 ng/µL, and 0.29 ng/µL for healthy controls.

To evaluate the quality of extracted cfDNA more precisely, the fraction of B-cell DNA in the samples was measured. We designed a duplex ddPCR assay—one dye measuring total genomic DNA (previously validated ddPCR assay) and other B-cell rearranged DNA. Some degree of B-cell contamination was observed in all samples. Mean value for CRC patients’ group was 0.69% ± 0.18%, for hemorrhoid patients’ group 1.57% ± 1.02%, and for healthy controls’ group 1.56% ± 0.88%. There was no statistical difference between the groups involved in our study (Figure 4). However, four samples were determined as outliers in the statistical analysis, and those were excluded from any further analysis.

### 3.3. Patients

Overall, 34 patients with locally advanced rectal cancer were included in the study. Matching tumor–serum samples were collected from these patients between 2012 and early 2015. The average age at the time of surgical removal of the primary tumor and blood withdrawal was 62 years old (min 42 years old, max 84 years old). The majority of tumors (72%) were located in the rectum with an average distance of 8 cm from the anal verge. According to Dukes classification, 7% of tumors were staged as Dukes A, 39% as Dukes B, and 53% as Dukes C (Table 1). Mucinous type of tumor was present in 25% of patients. At the time of diagnosis, none of the patients had confirmed distal metastasis.

### 3.4. Primary Tumor Mutation Status

We screened for the five most common mutations in colorectal cancer—KRAS G12A, KRAS G12D, KRAS G12V, KRAS G13D, and BRAF V600E. A total of 24 tumor samples were mutated (70.6%), and 10 samples were wild type (29.4%). The average fractional abundance of mutation in wild-type background was 9.50%, ranging from 0.009% to 38.9%. The most abundant mutation was KRAS G12D, which was present in 14 samples (41.18%). Other mutations were present in the following order: KRAS G12V in seven samples (20.59%), KRAS G13D in six samples (17.65%), KRAS G12A in one sample (3%), and BRAF V600E in one sample (3%). Among five samples harboring multiple mutations, one mutation was present in higher fractional abundance (more than 10%) and the other was minor (below 1%) present in four samples (Table 1).

### 3.5. Matching cfDNA Mutation Status with a Primary Tumor Sample

The average concentration of cfDNA was 0.29 ng/µL of serum ranging from 0.17 ng/µL to 38.9 ng/µL of serum. Serum samples were screened for the same mutations as primary tumors. Overall, 21 samples were wild type (61.76%), and 13 samples were mutated (38.24%). Four samples had multiple mutations—two samples had two mutations in low percentage (<1%), but at least one mutation was the same as found in the tumor. Two samples had three or more mutations of which one was also found in the matching tumor sample. Both of these multiple mutation serum samples also had a BRAF V600E mutation, which was a de novo mutation in both cases (Table 1). We compared the median fractional abundance of mutations in tumor and serum samples. For KRAS G12D, 11.16% versus 0.12%; for KRAS G12V, 9.60% versus 0.27%; for KRAS G12A, 19.9% versus 0%; for KRAS G13D, 5.34% versus 2.50%, and for BRAF V600E, 0.031% versus 0.32%. For KRAS mutations overall median fractional abundance for tissue was 9.84% and for serum 0.23%.

Because the overall frequency of mutations found in tumor samples was higher than in serum, we checked whether the presence of the mutation in serum was correlated with the percentage of mutation detected in the tumor sample. We did not find any correlation between the abundance of the mutations in the tumor and serum, suggesting that the higher fractional abundance of mutation measured in the tumor does not prejudice the higher possibility of mutation detection in serum (Figure 5).

### 3.6. Evaluation of Mutation Detection Specificity

The ddPCR mutation assays are probe-based, containing one probe binding to WT DNA and the other binding to MUT DNA. Because we were measuring only the most prevalent mutations in KRAS codons 12 and 13, there is always a possibility of the presence of some other mutation at this position, which would result in failed binding of both probes. In order to check for this possibility, we calculated the ratio between the measured WT DNA in BRAF assay and the mean value of all measured WT amounts of DNA in KRAS assays in tumor samples. WT measurements with KRAS assays are very similar because they probably contain identical WT probes. BRAF and KRAS WT probes bind to different locations, hence the estimation of WT DNA content should be unbiased. If we plot the fractional abundance of the most prevalent mutation on the same plot (Figure 6), we can see that with the higher ratio of BRAF/KRAS WT measurement, the fractional abundance of dominant detected mutation is higher. Three samples (9, 16, 31) seem to stand out. In two samples, we detected mutation at fractional abundance below 1% (0.29% and 0.07%), and the third is detected as WT. 

It was decided to look into these samples and performed Sanger sequencing following the cold PCR to detect any other possible mutations that could affect the binding of the KRAS WT probe. We did not find any additional mutations in KRAS at positions 12 and 13, but we found one sample harboring mutation c.59 C > T that was previously detected and registered in the COSMIC database (COSM4169161) [30].

## 4. Discussion

Because early detection is a critical goal in cancer screening, circulating nucleic acids represent a new class of biomarkers that could be used to improve the patients’ outcome [31]. The concept of ddPCR assay for the quantification of cfDNA is still not widely used. Several publications implicate the concentration of the cfDNA as a feasible diagnostic marker for malignancy detection as the amount of cfDNA is inevitably correlated with cancer progression [24]. But it is important to distinguish cell-free DNA (cfDNA) from circulating tumor DNA (ctDNA), which a lot of publications rely on. This expression can only be used if tracking a specific mutation previously found in a tumor, which is to ensure the origin of cfDNA and is used as a diagnostic marker, is known. There were attempts to distinguish patterns of the cancerous fraction of cfDNA [32], but this is not a widely accepted concept. cfDNA was measured in three groups—CRC patients and two control groups, i.e., patients with hemorrhoids and healthy controls. In this study, it was shown that a cfDNA measurement at only one time-point is not a useful diagnostic biomarker per se. In order to improve sensitivity and specificity required for extensive population screening for diagnosis of early-stage CRC, some authors propose the use of cfDNA concentration in a combination with protein biomarkers, such as carcinoembryonic antigen (CEA) [33]. In this case, the group of patients with hemorrhoids had elevated levels of cfDNA that partially overlapped with the concentration range of CRC patients. All kinds of inflammation processes in the body can influence the level of cfDNA. It can originate from tumor microenvironment cells and other healthy cells such as endothelial and immune cells. The localization of the cfDNA is limited to the tumor and can be derived from various parts of the body. It seems to be the case that all cells are likely to continuously release cell-specific DNA into the extracellular environment [34]. It was shown that severe injury such as blunt trauma and burns result in elevated levels of cfDNA [35], in addition to radiotherapy-induced infections [36], rheumatoid arthritis [37], or overtraining syndrome [38].

cfDNA is, in most cases, measured in blood plasma or serum. Therefore, sample processing has a great impact on the quality of cfDNA in terms of contamination with nucleic acids from nucleated blood cells [39]. cfDNA has a “ladder” pattern of ~150 bp and a smaller amount of its multiplications [32]. Researchers usually use capillary electrophoresis-derived bioanalyzers, which allow relatively precise measurements of fragment size [40]. The others measure the content of B-cell DNA fraction in the cfDNA [13]. We designed a duplex ddPCR assay that can quantify the cfDNA and measure the degree of B-cell DNA contamination in a single reaction. A certain degree of contamination with an average rate of 1.18% was measured, which indicates that the quantities of cfDNA are likely to be slightly overrated and that there is still room for improving sample processing procedures. On the other hand, none of the groups in our study was biased because of higher whole blood cell contamination. 

One of the aims of this study was to assess the use of ddPCR in the detection of somatic mutations in tumor samples of CRC patients. This group of patients was already analyzed with a different technique for mutation detection [25], and better sensitivity with ddPCR was observed. The mutations with a frequency of 5% or lower were previously not detected with the use of the SNaPshot Multiplex System. We checked whether the results detected with a more sensitive method still meet the expected frequencies. According to the COSMIC database, the most abundant mutation in the KRAS gene is G12D [30]. In adenocarcinoma samples from the large intestine, this mutation was present in 44% of all samples (results in this study: 42%). The second most common mutation was G12V, present in 28% of all cases (results in this study: 21%), then G13D in 24% (results in this study: 18%), and the least frequent among measured was G12A with 7% (results in this study: 3%). As a Serbian population was analyzed, these results were compared with a study performed within the same ethnic group [41] from a laboratory that performs routine diagnostic testing. The distribution of KRAS mutations was very similar—the most frequent mutations were G12D (44%), G12V (21%), G12A (10%), and G13D (10%). These results were also compared to the Slovenian population of CRC patients. The most common mutation is KRAS G12D (44%), followed by G12V (21%), G12A (10%), and G13D (10%) [27]. In the Serbian population, 17.8% of all KRAS WT samples were BRAF V600E mutated [41] and in the Slovenian population, this mutation was found in 5.1% of all KRAS WT cases [27]. In this study, mutation was found in only one tumor sample. The likeliest explanation for this discordant result is the fact that samples of locally advanced rectal cancer (LARC) patients were used, and a low percentage of BRAF mutations are expected [42]. Although this study of 34 patients with paired tumor/serum samples was relatively small, a much larger study performed on LARC patients [42] has similar numbers of paired samples, which makes this study appropriate for comparison. Sclafani et al. used ddPCR to measure three KRAS mutations from cfDNA in a single custom assay (G12D, G12V, and G13D) and compared the results with tissue mutation status measured with standard PCR techniques. Overall, 43% of patients had the same mutation status in the tumor and plasma, while 35% of patients with WT tumors were found to have a mutation in the plasma. They found that overall mutation status concordance was 56% and for specific mutation was 51% [42]. It was previously recognized that tissue-based genetic tests are limited by some inherent characteristics of cancer such as intra-tumor heterogeneity and clonal evolution [43], which makes cfDNA a more appealing medium for overall mutation analysis because it provides a better representation of the complete genetic landscape of a tumor [32]. In patients with lower stadiums of disease or where the disease is not metastatic, this should be appropriately validated. In this study, it was shown that overall median fractional abundance for KRAS mutations in cfDNA was approximately 50-times lower than in tumor DNA (0.23% versus 9.84%). The median mutation frequency detected in cfDNA is in accordance with the literature [42]. 

Liebs et al. performed a study where they measured the mutations in matching tumor/plasma samples and the highest accuracy (68%) was achieved in patients with distant metastases, demonstrating that cfDNA analysis in patients with earlier cancer stages is limited [23]. It also depends on the cancer type how prone the tumor is to release cfDNA into the bloodstream—the detectable levels of ctDNA can vary between tumor types even in patients in the metastatic phase of the disease. The concertation of ctDNA varies among patients, also in those with the same tumor type and stage of the disease [44]. Although colon cancer belongs to the group of cancers that release higher quantities of DNA into the bloodstream, Bettegowda et al. reported that in patients with localized disease (no clinical or radiographic evidence of distant metastasis at the time of sample collection) detectable levels of ctDNA are found in just 55% of patients. However, in this group of patients ctDNA was strongly correlated with the stage of the disease. 

For at least three patients (patients 2, 18, and 21), we found mutations in cfDNA but not in the tumor, which indicates the probability of sampling the tumor at a site without mutated lesions due to tumor heterogeneity. It was also noted that in two out of three cases the serum detected mutation was KRAS G13D. Previous reports showed that patients with RAS WT tumors and plasma that is positive for KRAS and BRAF mutations can be resistant to the EGFR inhibitor [20,21,44,45]. CRC presumably contains resistant mutant clones before treatment, and the proportion of these resistant clones increases under therapeutic pressure [46]. This indicates that mutation detection in liquid biopsy could in certain cases represent a more sensitive method for mutation detection than tumor tissue. In some studies, it was shown that the emergence of resistant KRAS MUT clones could be detected for up to 10 months before radiographic confirmation of disease progression [47].

We propose the supplemental approach similar to the study by Finzel et al. [48], where tissue and liquid biopsy were tested retrospectively on a set of patients with tumor stage IV who failed at least one line of therapy. They have shown that testing only tissue biopsy misdiagnosed a significant number of patients due to tumor heterogeneity. To use this approach and assess the diagnostic benefits in patients with the non-metastatic stage of the disease, further larger studies should be performed. The financial and life quality cost-benefit of the possible mistreated patients, who were not properly characterized because of tumor heterogeneity, should also be considered. In the usual clinical setting, just a fragment of the tumor is being used for the assessment of somatic RAS / RAF mutations because the pathology diagnosis should be set from the same tissue [49]. Although expensive, the testing of several tumor sites could be an option, but if the whole tumor mass is not used for this purpose, there will always be a question addressing the sensitivity because of tumor heterogeneity. cfDNA can be considered as a good option to supplement tissue biopsy testing and overcoming this problem as shown in other cancer types [50].

ddPCR is a tool that can quantify the number of amplifiable DNA molecules in a sample. Generally, when using mutation assays, only the ratio between WT and MUT assays is of interest to determine the percentage of mutations present. This study however wanted to see if more information from the WT assays could be extracted. Codons 12 and 13 of KRAS are hotspots for the mutations in several malignancies—lung [51], pancreatic [52], and colon cancer [53]. ddPCR mutation assays enable detection of discrete mutations, but could the absolute measurement of the WT probe provide us with the information about additional mutations not included in this analysis? BRAF WT probe information and normalized the average KRAS WT measurement was used. Three samples were found that did not follow the rule that with the increased percentage of KRAS mutations, the ratio of BRAF/KRAS increases. This could indicate the presence of an additional mutation in the KRAS region of codons 12 and 13. This could not be confirmed using Sanger sequencing, most likely because of the low sensitivity of the method. However, an additional mutation in KRAS codon 20, which may not have been covered by the probe in the ddPCR assay due to the distance from the codons 12, 13, was found. The detection of an additional mutation is most likely just a coincidence. A somewhat similar approach, but with a more refined concept, was used by Bidshahri et al. [54]. They developed a ddPCR assay using two BRAF WT probes for the detection of all BRAF V600 mutations based on a similar principle. This kind of assay could represent a highly sensitive method for ruling out all the WT patients in the first stage of testing.

## 5. Conclusions

In this manuscript, we describe a ddPCR assay that enables reliable and sensitive quantification of cfDNA and quality control assessment in terms of whole-blood contamination. The use of this assay in CRC patient management still needs to be evaluated. It could be used for monitoring the cancer reoccurrence after the primary operation or as a tool for cancer burden evaluation in metastatic patients. ddPCR offers the possibility of more sensitive detection of mutations in tumors’ tissue even in lesions with a minor number of mutated cells. With the additional testing of serum, we could overcome the problem of tumors’ heterogeneity because we can also detect the mutations that could be missed because of spatial sub-sampling of the tumor biopsy. Solely cfDNA mutation assessment cannot replace tumor tissue sampling for the purposes of mutation status detection in patients with non-metastatic colorectal cancer, but combined testing could provide more precise information.

## Figures and Tables

**Figure 1 genes-12-00289-f001:**
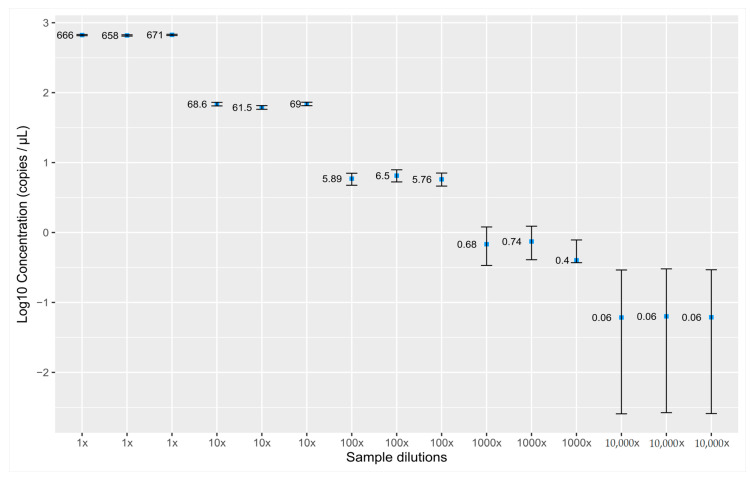
Linearity of the droplet digital PCR (ddPCR) assay for the quantification of cell-free DNA measured in triplicates in 10-fold dilutions over the five orders of magnitude. Dots represent measured concentration (copies/µL), error bars represent Poisson’s distribution coefficient of variation.

**Figure 2 genes-12-00289-f002:**
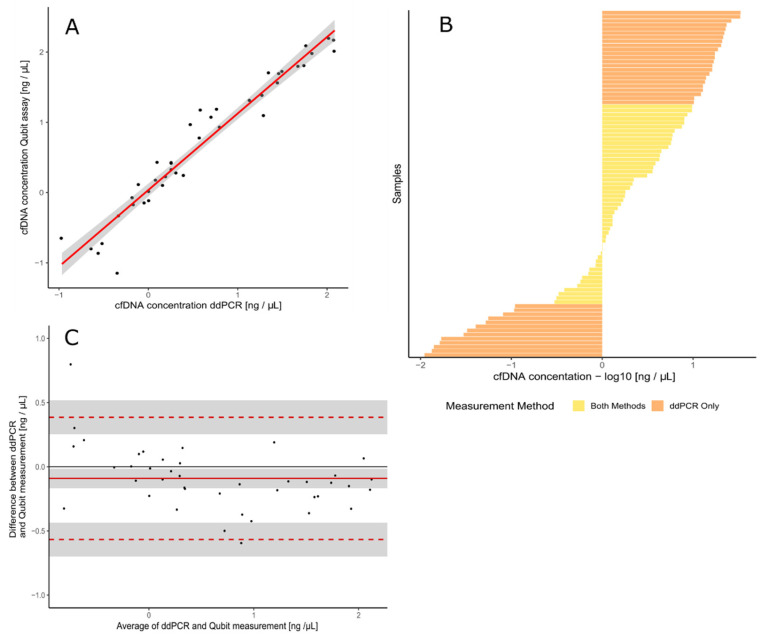
Comparison between ddPCR assay for quantification of cell-free DNA and Qubit. (**A**) ddPCR assay results show a good correlation with Qubit measurements (Persons’ coefficient of correlation r = 0.976)—the red line represents linear regression model line, and the grey area is 95% confidential interval of the linear regression model. (**B**) Dynamic range of ddPCR assay is much broader than Qubit-based measurement. (**C**) Bland–Altman difference plot representing high correlation and good agreement between the ddPCR and Qubit methods for samples of different concertation ranges.

**Figure 3 genes-12-00289-f003:**
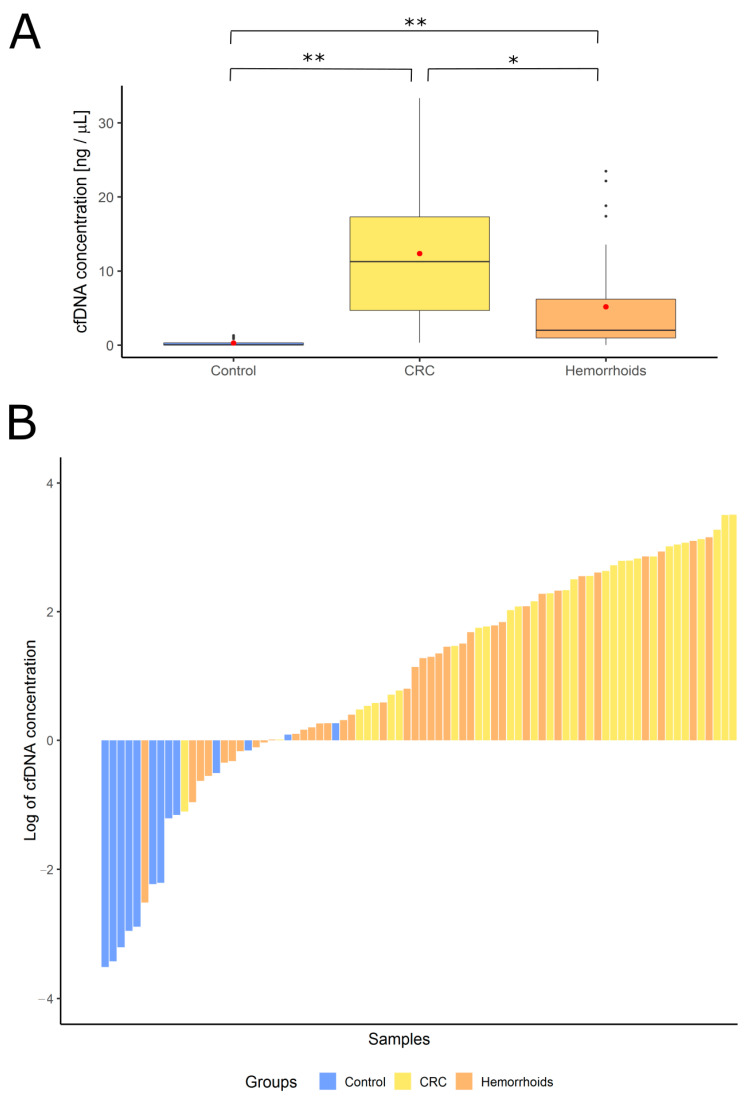
The concentration of cell-free DNA in different groups of patients. (**A**) Concentration ranges represented with box plots—red dots represent average values in each group (healthy controls 0.29 ng/µL, colorectal cancer patients 12.36 ng/µL and hemorrhoid patients 5.17 ng/µL). Statistical significance was calculated with *t*-test (* *p* < 0.05, ** *p* < 0.001). (**B**) Distribution of cfDNA concentration from lowest to the most concentrated sample based on the origin of the sample. Control—healthy controls, CRC—colorectal cancer patients, Hemorrhoids—patients with hemorrhoids.

**Figure 4 genes-12-00289-f004:**
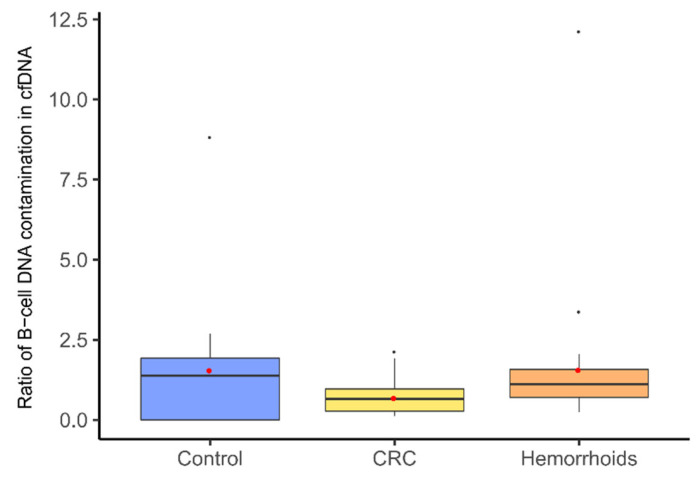
The ratio of B-cell DNA measured in cell-free DNA isolated from the serum of different groups of patients. Ratio ranges are represented with box plots—red dots represent average values in each group (healthy controls 1.55%, colorectal cancer patients 0.69%, and hemorrhoid patients 1.57%). One-way ANOVA did not show any statistically significant difference between groups. Control—healthy controls, CRC—colorectal cancer patients.

**Figure 5 genes-12-00289-f005:**
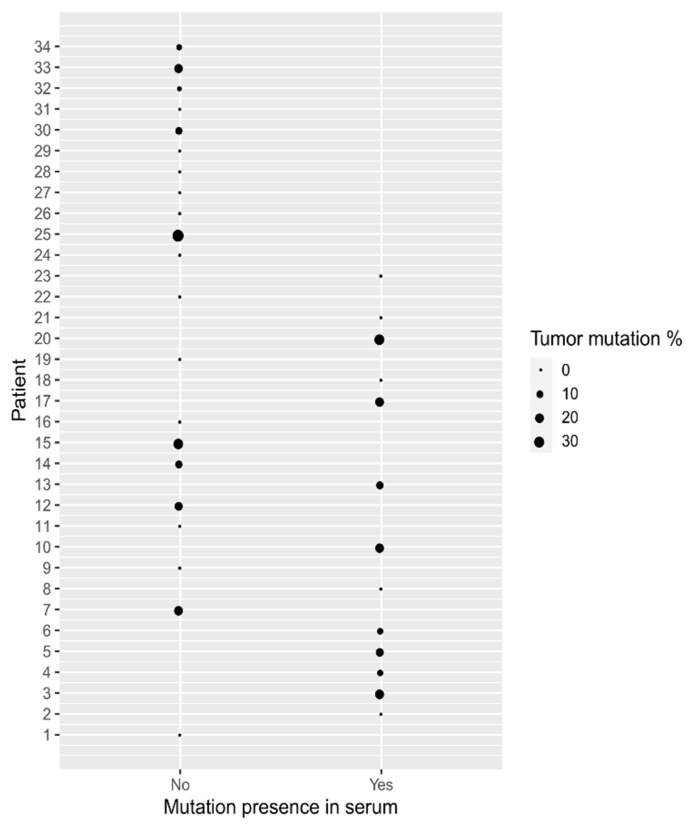
Graphical representation of the relation between the presence of mutation in serum and the percentage of mutation measured in the tumor. The Mann–Whitney U test did not show any correlation between a higher percentage of mutation in tumor and the chance that the mutation will be measured in serum as well.

**Figure 6 genes-12-00289-f006:**
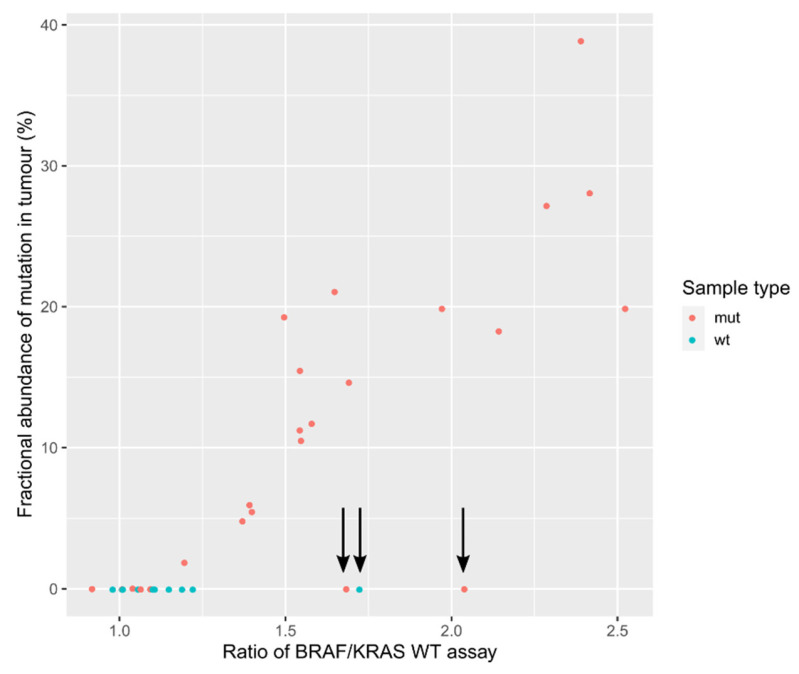
Plot representing the BRAF/KRAS WT assay measurement ratio and its dependence on the fractional abundance of measured mutation in tumor. If the sample is wild type (WT), the expected ratio of BRAF/KRAS WT assay measurement would be around 1. If the tumor sample is mutated (MUT), then a higher percentage of mutation is measured, increasing the BRAF/KRAS ratio. The three samples that do not follow the pattern are marked with arrows.

**Table 1 genes-12-00289-t001:** Overview of colorectal cancer patients’ samples used in tumor/serum matching study: classification of tumor samples according to Dukes, detected KRAS and BRAF V600E (*) mutations and their fractional abundance from tumor DNA, and cell-free DNA isolated from serum drawn just before surgery. ND—not determined, WT—wild type.

Patient No.	Dukes Classification	Tumor Mutations	Serum Mutations
1	B	WT	WT
2	C	WT	* V600E (0.13%)
3	ND	G12D (21.10%)	G12D (0.017%)
4	B	G12D (5.50%)	G12D (0.019%)
5	C	G12V (14.66%), G12D (0.23%)	G12D (0.18%), G12V (0.036%)
6	C	G12V (5.99%)	G13D (1.95%), G12D (0.08%), G12V (0.54%), * V600E (0.09%)
7	C	G12A (19.90%), G12D (0.26%)	WT
8	C	G12V (0.018%), G13D (0.014%)	G12V (0.07%), G12D (0.046%)
9	C	G12D (0.029%)	WT
10	B	G12D (19.30%)	G12D (0.25%)
11	C	WT	WT
12	B	G12D (15.50%), *V600E (0.031%)	WT
13	B	G13D (11.75%)	G12D (0.033%)
14	C	G12V (11.28%)	WT
15	B	G12D (27.20%), G13D (0.047%)	WT
16	A	G12D (0.027%)	WT
17	B	G12V (19.90%)	G13D (7.90%), G12V (0.43%), * V600E (1.00%)
18	ND	WT	G13D (0.11%)
19	C	WT	WT
20	B	G12D (28.10%)	G12D (0.31%)
21	B	WT	G13D (0.070%)
22	ND	G12D (0.009%)	WT
23	C	G12D (0.041%)	* V600E (0.60%)
24	C	G13D (0.017%)	WT
25	B	G12D (38.90%)	WT
26	C	G12D (0.065%)	WT
27	B	WT	WT
28	C	WT	WT
29	C	WT	WT
30	ND	G12V (10.54%)	WT
31	ND	WT	WT
32	C	G13D (1.90%)	WT
33	ND	G13D (18.30%)	WT
34	A	G12V (4.84%)	WT

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
