# Peer review of "Detection of Somatic Mutations with ddPCR from Liquid Biopsy of Colorectal Cancer Patients"

_genes, 2021, doi:10.3390/genes12020289_

Round 1

Reviewer 1 Report

The detection of mutations in the liquid biopsy samples of cancer patients holds great clinical relevance. It is important that assays designed to detect tumor mutations in liquid biopsy samples have high sensitivity and specificity. In this article, the authors have utilized the droplet digital PCR technique to detect somatic mutations in the serum of colorectal cancer patients, which has gained a lot of attention in recent years. Overall, the authors have performed a nice study.

Few suggestions and comments to the authors:

  1. Lines 46-58: This paragraph detailing the history of liquid biopsy is unnecessary and can be either shortened or removed.
  2. The authors should also provide patients characteristics belonging to healthy controls and hemorrhoids group (e.g., age, sex, etc.)
  3. Is there any information available on the tumor size of colorectal cancer patients? Bigger tumors tend to shred more DNA in the blood and could have impacted the cfDNA concentration measurements. It will be interesting if the authors could correlate the tumor size with the cfDNA DNA concentration.
  4. The discussion is too lengthy and should be shortened. 
  5. The authors should scan the article for spelling errors and correct them.

Author Response

We would like to thank the reviewer for hers/his thoughtful reading and helpful comments on the manuscript. We went through suggestions and here are our answers:

“Lines 46-58: This paragraph detailing the history of liquid biopsy is unnecessary and can be either shortened or removed.”

This paragraph is indeed obsolete for the matter on the article and was therefore removed from the introduction. It is not needed for the understanding of the scientific question addressed in the manuscript and we agree with the reviewer.

“The authors should also provide patients characteristics belonging to healthy controls and hemorrhoids group (e.g., age, sex, etc.)”

We included the data about the patients’ gender distribution for the case and control groups (lines 103, 107, and 109). The data for the age is not available for the hemorrhoid patients’ group and we could not obtain it, since the samples of this control group were collected in the same time period as CRC patients (2012 – 2015) and the age data was not recorded at the time of collection. We can assume, that given the fact, that these patients had surgical removal of hemorrhoids, they tend to be of older age. Regarding healthy controls, we have collected the clinical data about their health status: BMI, smoking status, co-morbidities, basic biochemical blood analysis, etc. and the age. Average age is 45 years (from 21 to 70). We could not find any correlation with the cfDNA concentration, so we consider that this data is not important for the interpretation of the results.

“Is there any information available on the tumor size of colorectal cancer patients? Bigger tumors tend to shred more DNA in the blood and could have impacted the cfDNA concentration measurements. It will be interesting if the authors could correlate the tumor size with the cfDNA DNA concentration.”

We thank the reviewer for this comment. We considered also that this information, in particular, could help us understand the wide range of concentrations measured in the CRC group. Unfortunately, these patients were recruited in the period from 2012 to 2015, and at the time of performing analysis for this manuscript, we could not obtain the information about the tumor size, since the records about pathological and clinical data are stored in different divisions in the hospital, often in the paper form. But of course, we will pay full attention to obtaining also the data for tumor size (and holding a digital database with clinical parameters for the patients enrolled) for any future studies.

“The discussion is too lengthy and should be shortened.“

We considered this remark and omitted all the sections of the discussion not crucial for the understanding of the matter discussed. We noticed that a sentence was mistakenly removed from the discussion when the review from co-authors was done. We inserted the accidentally removed sentence in lines 421 – 422.

“The authors should scan the article for spelling errors and correct them.”

The article was corrected for the spelling errors, as well as for the language style. We decided that passive form is more appropriate for the discussion. The meaning was not changed with these corrections.

Reviewer 2 Report

In this study, Zmrzljak et al. investigated the relevance of ddPCR assays for cfDNAs, which were customized for better quantification and quality control, in locally advanced colorectal cancers without metastasis (LARC). Although single utility of ddPCR assay for cfDNA seems less sensitive and specific for early detection of LARC due to minimal quantity of cfDNA and contamination in benign disease state, if combined with assays for biopsy samples, more accurate stratification of patients for EGFR mAb therapy might be accomplished because the issue of intra-tumor heterogeneity could be resolved with cfDNA assay in some cases. Further study is necessary to confirm the idea they presented. Specifically the competitors of their assays could be ones where multiple samples from different tumor location are tested. If only two sites are samples, the cost of the assays would be almost the same as the ones they presented. Better presentation of the utility of their assays is expected.

The manuscript is overally well written, but the introduction part should be revised. It is little ambiguous what the goal of study. The other parts are acceptable.

Author Response

We would like to thank the reviewer for hers/his nice words about the article. We addressed the issues raised by the reviewer in the comments below:

“Further study is necessary to confirm the idea they presented.”

We are aware that further studies would be necessary to assess the utility of the concept presented in this manuscript. Most appropriate would be an interventional study where patients would be treated according to mutation status determined with the proposed “combined” approach. But at this point, this is probably a bit too far-fetched plan, additional studies are needed to assess the background. We consider our study as a “small pilot study” checking the concordance of genetic profile in the primary tumor and serum. Particularly the sub-group of LARC patients is not often studied. Most studies deal with metastatic patients where a higher rate of concordance is expected. However, KRAS mutations can also be detected in exhaled breath condensate of healthy individuals [1] so careful interpretation is needed when stratifying patients eligible for the biological-drug therapy. We believe that our study represents a small stone in a paved way for the successful treatment of CRC.

“Specifically the competitors of their assays could be ones where multiple samples from different tumor location are tested. If only two sites are samples, the cost of the assays would be almost the same as the ones they presented. Better presentation of the utility of their assays is expected.”

We are not sure if we understood this part of the comment correctly. We are aware that any additional sample for the analysis represents an additional cost. But considering the heterogeneity of the tumor, we believe that the introduction of cfDNA analysis in the patient management would be a minor cost considering the savings for the miss-treated patient's management and care, not to mention patient well-being. An additional paragraph discussing this matter was introduced in the discussion section in lines  483 – 496.

“The manuscript is overally well written, but the introduction part should be revised. It is little ambiguous what the goal of study. The other parts are acceptable.”

We agree with the reviewer’s comment. The introduction section was revised and all parts not necessary for the background understanding were removed.

  1. Youssef, O.; Knuuttila, A.; Piirilä, P.; Böhling, T.; Sarhadi, V.; Knuutila, S. Presence of Cancer-Associated Mutations in Exhaled Breath Condensates of Healthy Individuals by next Generation Sequencing. Oncotarget 2017, 8, 18166–18176, doi:10.18632/oncotarget.15233.